# Citation-Similarity Relationships in Astrophysics Literature

**Nathaniel Imel**[1]     **Zachary Hafen**[1,2,3]
[1]University of California, Irvine     [2]Northwestern University     [3]Adler Planetarium
nimel@uci.edu
zachary.h.hafen@gmail.com

## Abstract

We report a novel observation about which scientific publications are cited more frequently: those that are more textually similar to pre-existing publications. Using bag-of-word document embeddings, we analyze quantitative trends for a large sample of publication abstracts in the field of astrophysics ($N \sim 300,000$). When new publications are ranked by how many similar publications already exist in their neighborhood, the median number of citations per year that the upper $50^{\text{th}}$ percentile receives is $\sim 1.6$ times the median of the lower $50^{\text{th}}$ percentile. When new publications are ranked by an alternative metric of dissimilarity to neighbors, the median citations per year that the upper $50^{\text{th}}$ percentile receives is $\sim 0.74$ times the median of the lower $50^{\text{th}}$ percentile. We discuss a number of hypotheses that could explain these citation-similarity relationships relevant to the science of science.

## 1 Introduction

Big data and modern computational infrastructure have made it possible for an interdisciplinary *science of science* to achieve remarkable progress in illuminating fundamental principles underlying aspects of contemporary science, including collaboration patterns, research novelty and problem selection, career dynamics, social bias, incentive structure, team size, and citation dynamics [Fortunato et al., 2018, and references therein]. The current study aims to contribute to the growing body of work on pressures shaping scientific research, introducing two simple and general metrics for empirically characterizing trends emerging from the semantic relationships between scientific documents.

Specifically, we explore how the similarity of new publications to prior research may relate to their subsequent citation rates. Obtaining bag-of-word document embeddings for the abstracts of a large sample of astrophysics articles, we use cosine similarity to approximate semantic similarity between documents, and to formulate two simple metrics for describing the local geometry around publications in this space. These metrics are (1) the *density* of a publication's existing surrounding region, which is proportional to the number of adjacent, pre-existing publications within a fixed distance in literature space, and (2) the *asymmetry* of a publication, which quantifies how 'far on the edge' a publication is with respect to its nearest neighbors.

We catalogue several empirical results. We find a positive relationship between the density of a publication's existing surrounding region and the citations it receives per year. We also find a negative global relationship between the asymmetry of a publication and the citations it receives per year. Intriguingly, however, citation rates can increase with asymmetry when research areas are below a critical maximum density. These preliminary results suggest that, with some interesting exceptions, publications that are more textually similar to pre-existing publications become cited more frequently.

37th Conference on Neural Information Processing Systems (NeurIPS 2023).

# 2 Quantifying emergent trends in scientific literature

## 2.1 Document embeddings

To construct the semantic[1] representation space of scientific literature, we represent each publication $i$ in our dataset as a high-dimensional word vector (i.e., document embedding) $\mathbf{p_i} \in \mathbb{R}^n$, so that the textual similarity between two publications can be approximated by cosine similarity. We obtained these document embeddings for each abstract using a bag-of-words approach, described briefly as follows. First, we preprocess all abstracts by removing every token that is not a noun, verb, or adjective, and then lemmatize and stem the remaining tokens. These tokens become the features of our document vectors. For each publication, the values of these features are just the number of occurrences of each token in its abstract. This results in vectors of roughly $10,000$ dimensions, depending on the sample of publications. We normalize all vectors to unity. While advances in the field of natural language processing support more sophisticated representations, to begin to explore the utility of our framework, we opt for a more simple encoding which is a computationally cheap (using sparse operations) measure of lexical overlap. Our embeddings are strongly correlated with a natural measure of text overlap ($\beta = 0.68$, $R^2 = 0.50$, $p \approx 0$, see Figure 4 in Appendix A.

## 2.2 Publication metrics

We now introduce two simple metrics inspired by methods from data science and computational physics. For each publication in our analysis, we compute measures of *density* and *asymmetry* with respect to the publication's $k$-nearest neighboring publications, which we refer to as the publication's *neighborhood*.

### 2.2.1 Density

We aim for density to quantify the extent to which a research area is already populated. It is defined as a constant number of $k$ publications divided by the minimum arc length enclosing all $k$ publications in the neighborhood. Formally, the density of a publication vector $\mathbf{p_i}$'s neighborhood is:

$$\rho(\mathbf{p_i}) = \frac{k}{\arccos \mathbf{p_i}^\top \mathbf{p_k}}, \tag{1}$$

where $k$ is a fixed number of neighboring publications to $\mathbf{p_i}$, and $\mathbf{p_k}$ is the $k$-th nearest neighbor. We norm all document vectors to be of length 1, which means the inner product between publication vectors in Eq. 1 is equivalent to their cosine similarity. Consequently, $\rho(\mathbf{p_i})$ has the intuitive interpretation that it quantifies the number of adjacent publications to $\mathbf{p_i}$ per radian.

### 2.2.2 Asymmetry

Asymmetry represents how much a publication is on the edge/fringes of a selected area of literature. We define the asymmetry of a publication to correspond to the magnitude of the net direction of the $k$-nearest publication vectors within the publication's neighbors. The asymmetry of a single publication $\mathbf{p_i}$ is defined:

$$\alpha(\mathbf{p_i}) = \frac{1}{k} \left\| \sum_{j=1}^{k} \frac{\mathbf{p_i} - \mathbf{p_j}}{|\mathbf{p_i} - \mathbf{p_j}|} \right\|, \tag{2}$$

where $\|\cdot\|$ is the Euclidean norm. If a publication has 0 asymmetry, it can be thought of as highly 'prototypical', being at the semantic center of its neighborhood.

## 2.3 Research question: relation to citation rate

Given these two metrics, which describe aspects of a publication's relationships to similar previous research, it is natural to ask how they might relate to *citations*. We assume that citations reflect

---

[1]Strictly speaking, bag-of-words vectors capture lexical, but not deeper semantic features of an article. For example, synonyms (incorrectly) lead to orthogonality between embeddings. We thank an anonymous reviewer for bringing this to our attention. Every subsequent use of 'semantic similarity' in this paper should therefore interpreted in this narrower sense of 'lexical overlap'. Future extensions will explore the effect of embeddings that have been shown to encode fine-grained semantic content.

fundamental currencies of scientific recognition, and that they can serve as an indicator (albeit a noisy one) of the degree of attention or interest in a topic for a scientific community. In measuring the relationships between these variables, we aim to quantitatively explore how the scientific recognition of new research may be predicted by its textual similarity to previous research.

We define the **citation rate** of a publication to be the total number of citations it has received until present, divided by the number of years since its entry date in ADS. [2] Note that citations are events that only occur after an article was published. In order to quantify a publication's similarity to *previous* publications, and also to ensure that any emergent trends in our metrics with citations do not trivially result as a matter of definition, we exclude all subsequently appearing publications from analysis when calculating the density and asymmetry for each publication.

## 2.4 Data

Our dataset was generated using the NASA Astrophysics Data System (ADS, Kurtz et al. 2000) because (i) access was free within reasonable rate limits and (ii) we limited ourselves to a field some of our authors are familiar with. ADS tracks the metadata of wide variety of publications in physics and astronomy, including all papers posted to the astrophysics section of arXiv (`astro-ph`). ADS keeps track of all such publications, as well as references and citations within them.

### 2.4.1 Collecting publications

We surveyed ten "regions" of the astrophysics literature. Each region consists of an initial, "core" publication, and over ten iterations of a similarity-based retrieval process, roughly $30,000$ publications are retrieved from ADS based on their similarity to this core publication. For each region, this was performed by projecting all retrieved publications into document-embedding space, and retrieving up to $4,000$ new publications that are referenced by or cite these embedded publications, ordered by their similarity to the initial core publication. Combining all ten regions, this resulted in a total initial pool of $303,229$ publications. This procedure is described in detail in Appendix B. The ten regions consist of one familiar paper to the authors and nine randomly chosen papers; see Appendix C for the list of these papers.

### 2.4.2 Convergence testing

To make our dataset more representative of the true distribution of available astrophysics articles, we consider only a subset of our initial sample of $303,229$ articles that has 'converged' with respect to our search procedure. We check for this convergence by tracking the identity of papers in the neighborhood of each publication as we iteratively add new papers to our sample. Selecting the size of this neighborhood —the number $k$ of nearest neighbors that must not change after multiple expansions —is a free parameter. There is a trade-off in selecting its value: with too few neighbors, density and asymmetry cease to be informative; requiring each neighborhood to include all data points results in $0$ total converged publications for analysis. We decided to select $k = 16$ neighbors per publication, which yielded $27,597$ total publications for analysis. For a visual depiction of the distribution of converged papers per neighborhood size), see Figure 5 in Appendix B.1.

## 3 Results

We compute density and asymmetry for the document embedding of each publication in our sample of $27,597$ astrophysics articles from ADS, as well as the citations per year for each publication. Analyzing the relationships between these variables yields several emergent trends, which are depicted in Figure 1. If publications are ordered by the density of their neighborhoods, from least to most dense, then number of citations per year that the upper $50^{\text{th}}$ percentile receives is a median $\sim 1.6$ times that of the lower $50^{\text{th}}$ percentile. Second, if publications are ordered from least asymmetric (i.e., very prototypical, at the 'semantic center' of their neighborhood) to most asymmetric (furthest from the center), then the number of citations per year that the upper $50^{\text{th}}$ percentile receives is a median $\sim 0.74$ times that of the lower $50^{\text{th}}$ percentile. Least-squares regressions showed that density

---

[2]Future work may explore adjusted measures of citation rate; for example, excluding some years after publication and before the present year, and using the median number of citations each year, rather than our average, which is more sensitive to outliers.

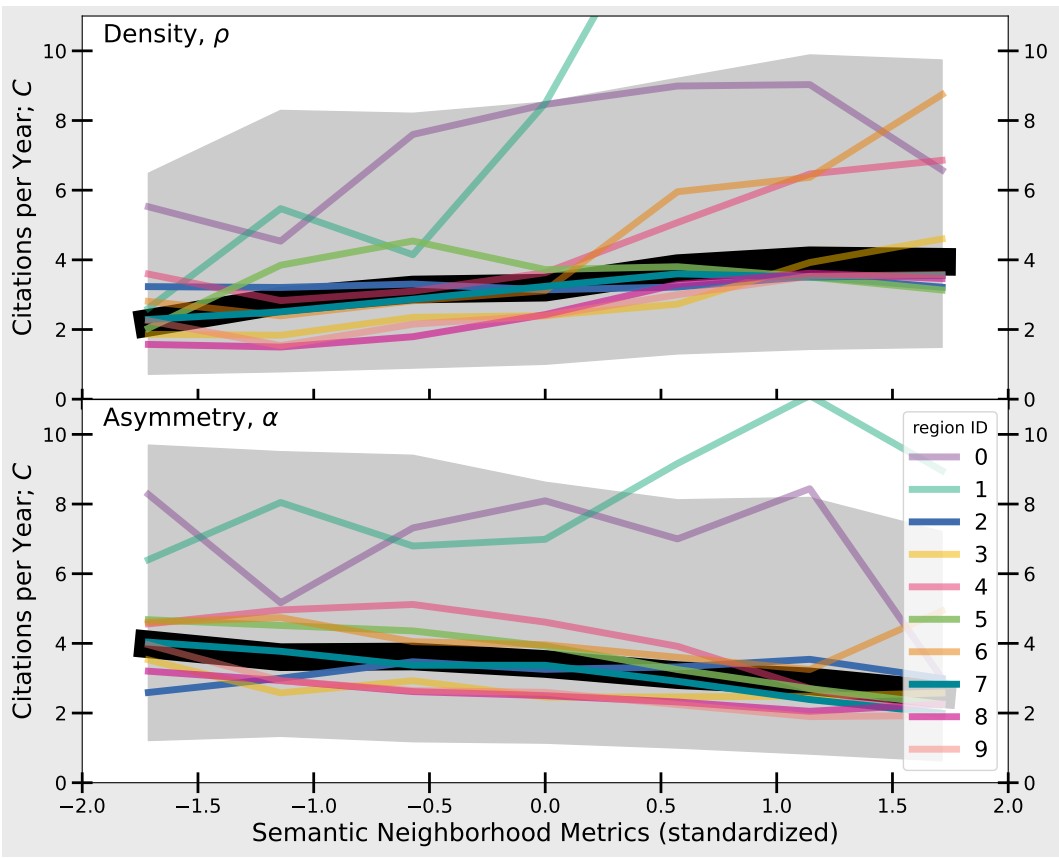

Figure 1: Trends for publication citation rates as a function of two metrics quantifying semantic relationships to previously existing publications. Data plotted correspond to **27, 597 astrophysics abstracts**, with metrics computed with respect to each of their nearest 16 neighbors in document-embedding space. **x-axis**: values of our metrics, standardized to zero mean and unit variance. **y-axis**: citations per year of the astrophysics publications (not standardized). **Density** $\rho$ represents how 'populated' the region surrounding a publication is, via the number of adjacent publications per radian (Eq. 1). **Asymmetry** $\alpha$ measures how 'on the edge' a publication is from the center of its neighborhood, via the average of magnitude of semantic difference (Eq. 2). Black line: running median line for all regions. Colored lines: running medians for different **regions** of astrophysics literature surveyed, generated by choosing a random astrophysics publication and iteratively retrieving similar ones (see Section 2.4.1). Gray region: 16th-84th percentile of distribution for combined regions.

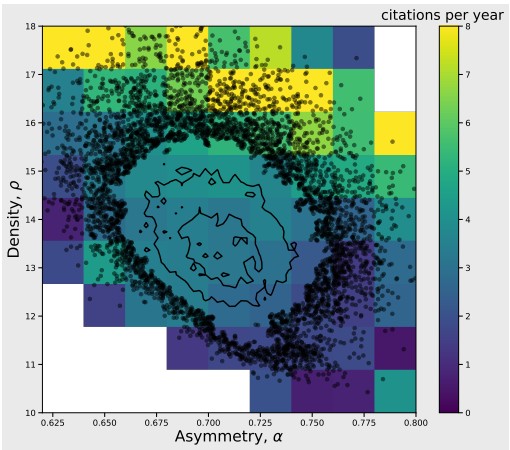

Figure 2: Relationship between density and asymmetry (not standardized) with respect to citation frequency. Each point represents a publication; the center of the distribution is visualized with contours to reduce overplotting. Color indicates the average number of citations a publication received per year (see Section 2.3).

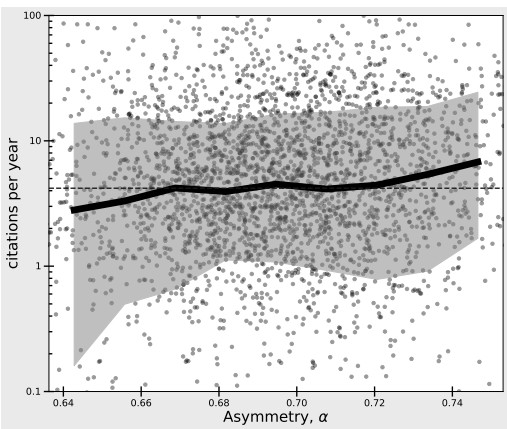

Figure 3: Asymmetry vs. citation frequency for publications with density values between $\rho = 15$ to $\rho = 17$ adjacent publications per radian. Each point represents a publication; the dashed line indicates the global median citations per year; the black line shows the running median; gray region delimits the $16^{\text{th}}$ and $84^{\text{th}}$ percentiles of the distribution.

($\beta = 2.6$, 95% CI $= [1.7, 3.4]$, $p < 7.4 \times 10^{-10}$, $R^2 = 1.4 \times 10^{-3}$) and asymmetry ($\beta = -65.2$, 95% CI $= [-101.0, -29.4]$, $p < 3.5 \times 10^{-4}$, $R^2 = 4.6 \times 10^{-4}$) are significant predictors of citations per year, although they cannot alone explain the high variability in citation frequencies. This variability is expected, given that citation dynamics reflect highly complex processes. We also compared density and asymmetry to each other to see how they might interact to predict citation rates (Figure 2). For all values of asymmetry, citation rates tend to increase overall with density. Restricting to just publications in the highest density areas ($\rho = 18$), we observe that citation rates tend to decrease with asymmetry. Intriguingly, for publications in moderately high dense areas ($\rho = 15$ to $\rho = 17$), citations tend to increase in asymmetry (see Figure 3).

## 4 Discussion

The overall relationships that density and asymmetry have with citation rates suggest that scientific publications that are more textually similar to pre-existing publications tend to be cited frequently. There is large variance in these trends, however, suggesting that density and asymmetry alone are poor predictors of citation rates. This variance may come from many factors; most obviously, other causes of citations that are not captured by our similarity-driven metrics, such as scientific and economic value. Over and above these factors, we expect that variables like author seniority, institution, and journal prestige can cause very similar papers to receive very different citation rates. Importantly, the overall picture that similarity to previous research tends to result in more citations is complicated by the fact that when research topics reach certain values of density, citation rates increase with asymmetry. Future work should investigate this trend more closely.

Still, it remains to be explained why we observe the citation-similarity relationships at all. One explanation might be that papers with similar language to existing ones are easier to identify. The prior density and centrality of a paper's topic, for example, may simply indicate the number of researchers who can possibly discover and subsequently cite the paper. Given that scientific output is growing exponentially Bornmann and Mutz [2015], we believe it is plausible that these observed trends reflect the outcome of resource-constrained exploration of research problems.

Another salient possibility is that density and asymmetry capture the dynamics of tradition and innovation in science. For example, if novel or innovative research is captured by highly asymmetric papers in sparse regions, and traditional, more conservative science corresponds to papers located at the center of densely populated topics, then the trends we observe would lend additional support to

the idea of an 'essential tension' shaping science Kuhn [1959], Bourdieu [1975]. There is significant quantitative evidence for such a tension between productive, conservative science and risky, innovative breakthroughs Foster et al. [2015]. Meanwhile, Yin et al. [2023] find that when using neural-network based document embeddings, the cosine dissimilarity of a new paper to existing papers correlates with validated measures of scientific novelty. Under the assumption that our metrics reflect dimensions of tradition and innovation, it is interesting that increasing asymmetry tends to yield higher citations at moderate levels of research density, as we speculate this could indicate that there is selection pressure for certain kinds of innovation in science, but only before a topic becomes too populated.

There are several important directions to extend the current study. The observed trends should be checked for robustness across diverse scientific fields and larger scales, as permitted by e.g. Lo et al. [2020], Garfield [1955]. Additionally, a comparison of results should be performed using multiple contemporary methods for embedding scientific documents using state-of-the-art language models (i.e. SciBERT, GPT-4, etc.), which are likely to represent not just textual similarity but richer dimensions of conceptual relatedness Beltagy et al. [2019], OpenAI [2023]. Beyond replication, our simple metrics provide a general framework to investigate research dynamics in novel ways. For example, how do research density and asymmetry evolve over time, with respect to particular topics, collaborations, and institutions? Can these metrics be refined to more precisely test hypotheses about conservative vs. innovative science? Can they be used to create artifacts that are useful to practicing scientists? We look forward to exploring these questions in future work.

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

# A    Obtaining document embeddings

Here we describe the vectorization process we performed for each publication.

Using the freely available API `https://ui.adsabs.harvard.edu/help/api/` and its associated Python package `https://ads.readthedocs.io/en/latest/`, we obtain abstracts and metadata for multiple publications at once.

To preprocess the abstracts, we use the default tokenizer from the Natural Language ToolKit (NLTK, Bird et al. [2009]) on the text of each abstract to remove stop words and all tokens that are not nouns, verbs or adjectives. This filters out high-frequency, uninformative words, but also numerical and mathematical content, both of which are common in astrophysics abstracts. We then stem and lemmatize the resulting list of tokens.

We then obtain the frequency counts of individual post-processed items. For an abstract with $n$ total unique items, we obtain a feature vector $\mathbf{p} \in \mathbb{R}^n$ by assigning each feature $f_i$ the frequency of the $i$th item in the abstract.

Since different abstracts will contain items not present in others, before using our vectors for similarity-based retrieval or analysis, we ensure that all vectors in a sample of abstracts have the same dimension by taking the union of all of their features, and then performing the frequency-based vectorization process described above.

## A.1    Correlation with text overlap

As a preliminary analysis of the quality of our simple bag-of-word vectors, we examined whether cosine similarity between two document vectors was interpretable as a proxy for mere text overlap. To investigate this, we randomly sampled $496,506$ astrophysics publications from ADS acccording to the procedure described in Appendix C, and computed the pairwise cosine similarity and text overlap between each document in this sample. We define the text overlap between two documents $A$ and $B$ as

$$\mathrm{overlap}(A, B) = \frac{n_{\mathrm{shared}}}{N}, \tag{3}$$

where $N$ is a normalization factor representing the total number of post-processed items between the two documents, which we define as the geometric mean of the number of total items in each documents, $|A|, |B|$:

$$N = \sqrt{|A||B|}. \tag{4}$$

The number of shared items between $A$ and $B$ is

$$n_{\mathrm{shared}} = \sum_{w \in W} \min(\text{count of } w \text{ in } A, \text{count of } w \text{ in } B) \tag{5}$$

where $W$ is the union of all the post-processed items in $A$ with all of the post-processed items in $B$.

An OLS regression revealed that cosine similarity was strongly correlated with textual overlap ($\beta = 0.68$, $R^2 = 0.50$, $p \approx 0$); this fit is depicted in Figure 4.

# B    Data retrieval loop

Here we describe the procedure by which we iteratively built out the ten regions of literature used in our analysis. Given each initial publication 'center' (i.e., one of the ten publications listed in Table 1), we performed the following:

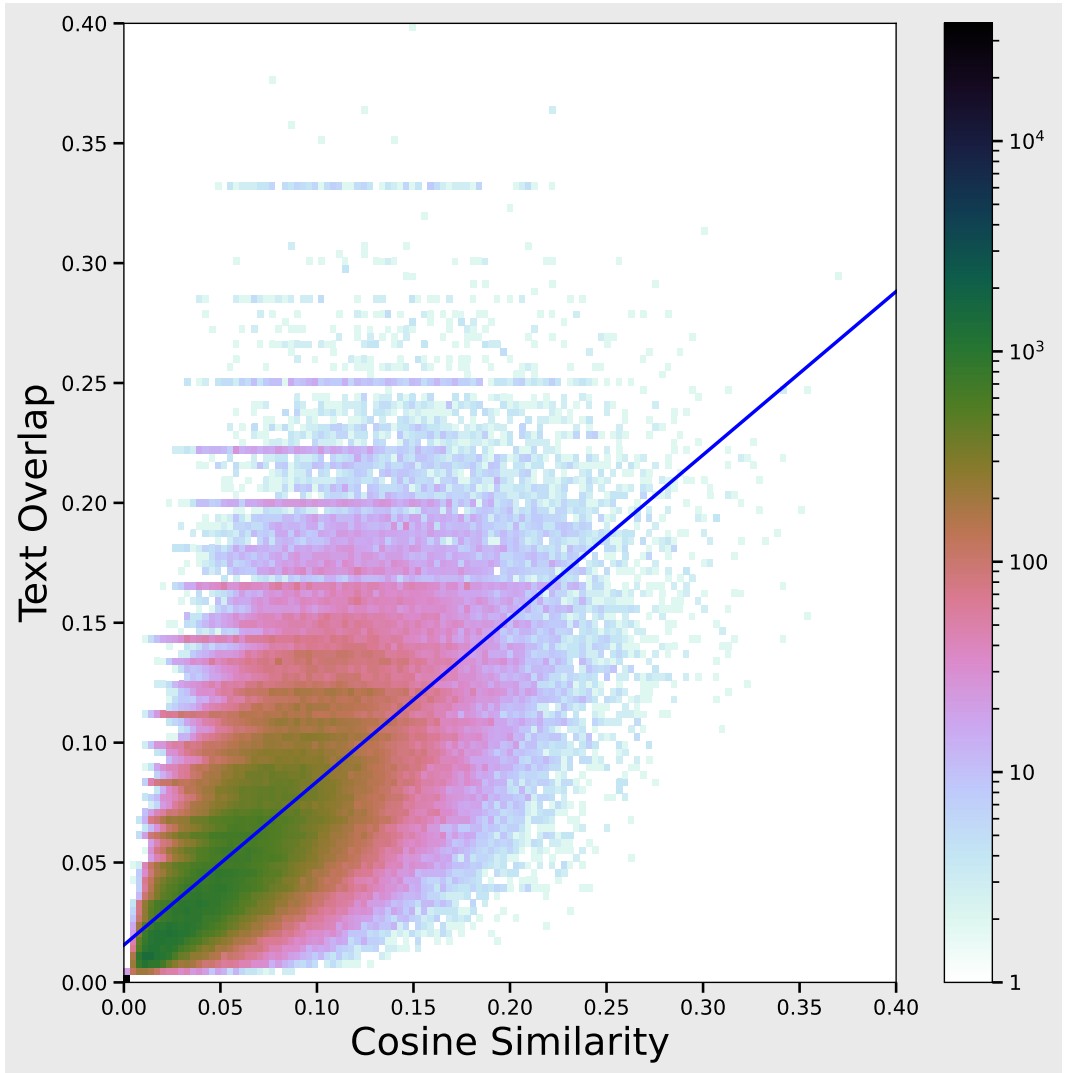

Figure 4: Cosine similarity vs. text overlap for each pair of abstracts in a sample of $496,506$ randomly chosen astrophysics publications. Color indicates counts of the distribution. The blue line indicates the line of best fit.

1. Include in the region all publications that cite, or are referenced by, the central publication (by importing them from ADS).

2. Project all publications in the region as document embeddings, and calculate the angle $\Psi$ between each publication and the central publication.

3. Designate all publications that cite, or are referenced by, the publication with the smallest $\Psi$ for inclusion in the region.

4. Repeat step 3), mutatis mutandis, for the publication with the second smallest $\Psi$, then the third smallest $\Psi$, etc., until between $1000 - 4000$ new publications are designated for inclusion to the region.

5. Import the designated publications from ADS, and include them in the region.

6. Repeat steps 2) - 5) nine additional times, for a total of ten expansions of the region.

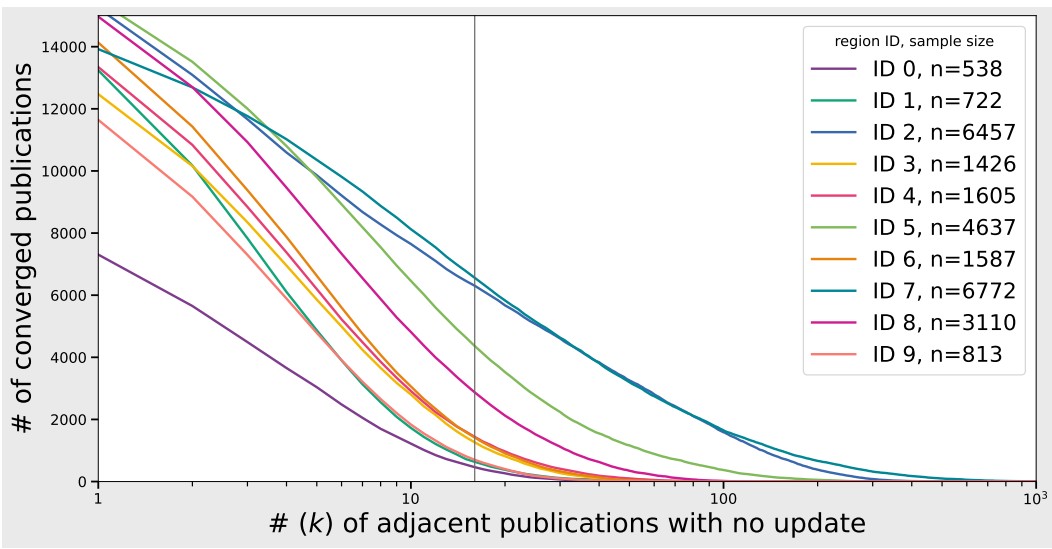

Figure 5: Within our dataset of $250,000$ total publications, we measured the number of publications that were not affected by additional searches for new related publications. $x$-axis: the number $k$ of nearest neighbors to each publication. $y$-axis: the total number of publications in our dataset such that each does not experience a change in its $k$ nearest neighbors after $3$ consecutive iterations of our search procedure. A black line is plotted at $k = 16$; its intersections with each region are the number of converged publications, indicated in the legend.

## B.1 Analysis of publication neighborhood vs. convergence

Figure 5 depicts the number of converged publications for analysis for neighborhood sizes $0$ to $1000$, by region. The black line indicates our choice of $k = 16$ as the neighborhood size for our calculations of density and asymmetry. All publications that were not converged were excluded from analysis.

## C Initial publications

To select the initial, 'seed' publications for our ten regions (Table 1), we first choose a random date, and randomly select a publication from the publications added to ADS on that date. If there are no suitable publications we choose another random date and re-peat this process. To select for astrophysics publications we filter to the arXiv classes `astro-ph.GA`, `astro-ph.CO`, `astro-ph.EP`, `astro-ph.HE`, `astro-ph.IM`, `astro-ph.SR`. We also filter on time by limiting our selections to between 1996 and 2018. Earlier than 1996, it become increasingly likely that a date will not have new entries added, making the aforementioned method of selecting random publications inefficient. We choose 2018 to allow for three full years to have passed from entry date to time of analysis.

| Title | Reference |
|-------|-----------|
| Low-redshift Lyman limit systems as diagnostics of cosmological inflows and outflows | Hafen et al. [2017] |
| Evidences for Collisional Dark Matter In Galaxies? | Salucci and Turini [2017] |
| Cosmological tests of an axiverse-inspired quintessence field | Emami et al. [2016] |
| Data Selection Criteria for Spectroscopic Measurements of Neutron Star Radii with X-ray Bursts | Ozel et al. [2015] |
| Magnetically-induced outflows from binary neutron star merger remnants | Siegel and Ciolfi [2015] |
| 100 deg$^2$ Mock Galaxy Cone for HI Surveys with the Early SKA | Obreschkow and Meyer [2014] |
| The Cosmological Parameters 2010 | Lahav and Liddle [2010] |
| Remarks on Statistical Properties of the Turbulent Interstellar Medium | de Avillez and Breitschwerdt [2006] |
| Analysis of astronomical data from optical superconducting tunnel junctions | de Bruijne et al. [2002] |
| Instabilities and Clumping in Type IA Supernova Remnants | Wang and Chevalier [2001] |

Table 1: The initial publications for our ten different regions of astrophysics literature.

