# OpenReview forum: "Citation-Similarity Relationships in Astrophysics Literature"
_NeurIPS.cc/2023/Workshop/AI4Science — NeurIPS2023-AI4Science Poster_

### Official Review · Reviewer_k4BD · 2023-10-10
**Interesting paper, problematic methods.**

**Rating:** 5
**Confidence:** 4

**Review:**

The authors study the relationships between the citation of an article and its similarities to other articles. For this, the authors use BoW as the article representation and quantitatively analyze the correlations between three article-level variables: (1) density; (2) asymmetry; (3) citation. (1) and (2) are both inter-article similarity metrics newly defined in the study. This paper finds that articles with higher density tend to have higher citations, which indicates articles that are more textually similar to other articles are more likely to be cited. Although the experimental design and the findings are quite interesting, there are some technical flaws that need to be addressed:

1. **Article representation**. The authors use BoW for representing articles, which only captures the *lexical* but not the *semantic* features of an article. For example, synonyms are (wrongly) treated as orthogonal by these metrics. As such, the use of *semantic* in the article is incorrect. BoW also brings more computational burden if not implemented as sparse operations. I recommend the authors use SciBERT-derived representations for the analysis. I also recommend that the authors visualize the article representations in a 2D figure (e.g., by t-SNE) so that the readers can have a much better understanding of the clusters.
2. **Density definition**. It seems that when the authors calculate the density of one article, they consider both the articles published before it and after it. If so, this is problematic because articles with high citations will have more similar articles (assuming that the citation articles are similar to the seed article). As such, if density is defined this way, "those that are more textually similar to pre-existing publications." is not true because you are considering both pre-existing and post-existing articles. That makes the finding much less useful.
3. **No case study**. The authors state that one reason for choosing to analyze astrophysics papers is because they are domain experts. Given this wonderful advantage, I feel the paper would be much stronger if they further confirmed their quantitative findings by manually studying a small subset of the dataset. Manual annotations by domain experts would be an impactful contribution to science of science.

Some minor notes:
- It seems like the study was conducted in 2021, and this paper would be much stronger if it was updated for 2023.
- Authors only spend a few sentences on related works. It would be great to see more elaboration on how this study relates to other prior work in science of science.
- On a more minor note, the findings are quite interesting but it'd be even more compelling if authors could elaborate more on the implications for the field, and perhaps even suggest actionable recommendations for other researchers.

---

### Official Review · Reviewer_LDYH · 2023-10-22
**Interesting work and ideas merging computational social science and physics**

**Rating:** 7
**Confidence:** 5

**Review:**

The authors present an interdisciplinary study integrating ML and computational sociology, examining 300,000 scientific papers within the field of astronomy. Specifically, their objective is to establish a relationship between paper similarity and their respective citation rates. To achieve this, the authors introduce two measures, interpreted as density and asymmetry.

To establish this relationship, the authors employed word2vec encoding to convert nouns, verbs, and adjectives from paper abstracts into a high-dimensional space. This approach allows them to compute the distances between individual papers.

The authors find a weak correlation between similarity and the citation rate. The correlation comes from very hand-selected data points (chosen from topics known to the authors). The correlations are very noisy and have a large spread. The authors provide multiple explanations for this outcome. It is an interesting indication of a correlation, but I would expect a much more careful investigation in subsequent peer-reviewed journal publications.

Reasons for accepting the paper:

-> The paper is well-structured and accessible, allowing even non-experts to comprehend its contents with ease.
-> It delves into an intriguing intersection of disciplines that hasn't been well-represented in past workshops, notably the computational social sciences.

Concern:
-> The main outcome of the paper is gained through a very small dataset of 10 points (hand-selected by the authors). The authors find some weak statistical correlations, but I wonder whether they remain when unbiased data points are taken into account. I think much more effort will be necessary for a submission to a peer-reviewed journal, and this workshop contribution can be seen as a very early first potentially promising indication of a correlation.

Additional Feedback:
The paper's references are not current. Apart from a mention of GPT-4, the most recent citation dates back to 2020 (marking the introduction of a relevant database). There exists a wealth of recent research in related areas that the authors touch upon, specifically in modelling and predicting citations. The absence of these references is noticeable. A "Related Works" section would provide a much-needed context, enabling readers to better appreciate the novel aspects of this study in relation to existing literature.

---

### Meta-Review · Area_Chair_KbhF · 2023-10-26

**Recommendation:** Accept (Poster)
**Confidence:** 3

**Metareview:**

This paper explores the similarities of papers in astronomy and their relationship to citation rates. The reviewers have found that the work is interesting but they raised some valid concerns. The authors should take these concerns seriously as it could help with the presentation of their work. Recommendation: Poster.